# Understanding Complex Relationships between Human Well-Being and Land Use Change in Mozambique Using a Multi-Scale Participatory Scenario Planning Process

Pedro Zorrilla-Miras [1,*], Estrella López-Moya [1], Marc J. Metzger [2], Genevieve Patenaude [2], Almeida Sitoe [3], Mansour Mahamane [4], Sá Nogueira Lisboa [3,5], James S. Paterson [2] and Elena López-Gunn [1]

1  I-Catalist S.L. C/Borni 20, Las Rozas, 28232 Madrid, Spain; elopezmoya@icatalist.eu (E.L.-M.); elopezgunn@icatalist.eu (E.L.-G.)
2  School of GeoSciences, The University of Edinburgh, Drummond Street, Edinburgh EH8 9XP, UK; marc.metzger@ed.ac.uk (M.J.M.); Genevieve.Patenaude@ed.ac.uk (G.P.); james.paterson@ed.ac.uk (J.S.P.)
3  Faculty of Agronomy and Forest Engineering, Eduardo Mondlane University, Maputo P.O. Box 257, Mozambique; almeidasitoe@gmail.com (A.S.); sanogueiralisboa@gmail.com (S.N.L.)
4  Centre Régional AGRHYMET, Université de Diffa, Niamey BP 11011, Niger; msourtchiani77@gmail.com
5  N'Lab, Nitidae, Agostinho Neto Avenue, Maputo P.O. Box 679, Mozambique
*  Correspondence: pzorrilla-miras@icatalist.eu; Tel.: +34-622-644-620

**Abstract:** The path for bringing millions of people out of poverty in Africa is likely to coincide with important changes in land use and land cover (LULC). Envisioning the different possible pathways for agricultural, economic and social development, and their implications for changes in LULC, ecosystem services and society well-being, will improve policy-making. This paper presents a case that uses a multi-scale participatory scenario planning method to facilitate the understanding of the complex interactions between LULC change and the wellbeing of the rural population and their possible future evolution in Mozambique up to 2035. Key drivers of change were identified: the empowerment of civil society, the effective application of legislation and changes in rural technologies (e.g., information and communications technologies and renewable energy sources). Three scenarios were constructed: one characterized by the government promoting large investments; a second scenario characterized by the increase in local community power and public policies to promote small and medium enterprises; and a third, intermediate scenario. All three scenarios highlight qualitative large LULC changes, either driven by large companies or by small and medium scale farmers. The scenarios have different impact in wellbeing and equity, the first one implying a higher rural to urban area migration. The results also show that the effective application of the law can produce different results, from assuring large international investments to assuring the improvement of social services like education, health care and extension services. Successful application of these policies, both for biodiversity and ecosystem services protection, and for the social services needed to improve the well-being of the Mozambican rural population, will have to overcome significant barriers.

**Keywords:** multi-scale scenarios; participatory scenario planning; social-ecological system; poverty alleviation; land use change; nature's contributions to people; Mozambique

**Research Highlights:**

1.  An increase in LULC change in Mozambique for 2035 is projected by all scenarios
2.  The most important drivers are social empowerment and effective law application
3.  Biggest differences stressed by policies that promote small or large-scale agriculture
4.  The multi-scale approach reveals hidden differences in local economies
5.  The participatory approach can be valuable to use in other least developed countries

## 1. Introduction

The successful transition towards a global society without extreme poverty by 2030 is one of the main objectives of the Sustainable Development Goals [1]. This transition

should occur as part of a wider global shift to ensure human development occurs within planetary boundaries [2]). Changes in land use and land cover (LULC) are one of the principal drivers for the degradation of nature [3]. LULC change is currently causing the loss of 13 million hectares of forests every year and 12 million hectares of arable land are being degraded annually, affecting an estimated 1.5 billion people globally with a disproportionate amount (74%) hitting the poorest and most vulnerable [4]. The provision of many ecosystem services (ES) depend on land, so future LULC changes and land degradation will affect poor populations disproportionally, especially those compounded by a lack of alternatives [5,6].

Globally, an estimated 767 million people live in extreme poverty, with 42% living in the Sub-Saharan Africa region [7]. The majority (about 80%) live in rural areas and 64% work in agriculture. Poverty is a complex concept and there is not an international consensus on its definition; however, we understand poverty as the inability to meet minimum standards and functioning, such as access to clean drinking water and sanitation or having a minimum level of formal education [8]. Rural dwellers face greater difficulties in achieving some of those standards, such as, for example, reading, access to electricity or use of safely managed drinking water [7]. On the other hand, rural dwellers have access to a higher diversity of ecosystem services than urban residents.

Rural inhabitants rely on ecosystem services in many different ways: provisioning services for obtaining wood products for construction, tools and fuel; food like fruits, hunting animals, mushrooms, etc.; grass for livestock; regulating services such as a good quality water, land for agriculture, and climate services; and cultural services, like access to sacred places or areas for recreation [9,10]. In some cases, these ES help rural families as a coping strategy in critical situations or contribute to poverty alleviation [11,12]. Therefore, changes in wellbeing are expected to occur if the integrity of ecosystems providing essential ES for the poor are degraded [3,13,14].

Because of increasingly complex and unpredictable global circumstances, as well as a growing understanding of socio-ecological systems, land management and land use problems require solutions that acknowledge and manage uncertainty [15]. Reversing the trends of LULC degradation and promoting a sustainable poverty reduction strategy requires a deeper knowledge of the complex processes that drive and link LULC change and poverty reduction. There is a growing understanding that land use systems are dynamic and connected across scales [16,17], as well as the social, economic and environmental factors affecting poverty. Each region is affected differently by a wide range of drivers, which in turn are shaped by a globalised economy. Better knowledge and understanding of these multi-scale relationships will facilitate the provision of more realistic and holistic governance strategies [18,19]. For example, Butler et al. [20] found that stakeholders at higher levels proposed more transformative strategies than local stakeholders. Therefore, new insights into the relationships between LULC changes and the multiple dimensions of well-being require the examination of interrelations and interdependences between ecological and social systems across scales [21]. Radical transformations are needed for reaching long-term sustainable social-ecological systems, and creative and experimental approaches are needed to envision and conceptualize them [22,23].

The inclusion of stakeholders in research is a way to deal with uncertainty and bring science closer to the problems faced by managers and local communities [24,25]. This co-production approach is increasingly being used to produce useful research for decision-makers with complex, long-term and large-scale challenges [26–28]. One of the tools that is increasingly being used for co-creation, dealing with future uncertainties and creative thinking is participatory scenario planning, which has gained popularity in recent years [29–32]. The development of scenarios is a methodology frequently used to analyse the complex processes that drive changes in LULC [33–35]. Indeed, many researchers see the necessity in developing participatory scenarios processes across-scales to support transformational changes, to compare results and to better understand how cross-scale interaction will affect future societies [36–38]. In this study scenarios are considered



descriptions of different plausible future situations that consider the uncertainty that exists beneath complex interactions of multiple factors. These scenarios are not predictions or forecasts, because they do not identify the most probable future [39,40]. In their simplest form, scenarios can be a vision for the future which can prepare individuals, communities and institutions for uncertainty and complexity through social learning [41], stimulating discussion and creative thinking [42].

Participatory scenarios have been used to study LULC changes in sub-Saharan Africa at regional [21,43–45] and local scales [19,45–49]; and there are several scenario exercises developed at multiple scales simultaneously [50–54]. Nevertheless, to the best of our knowledge, there is only one study that developed scenario exercises at multiple scales in sub-Saharan Africa [17]. There are also very few examples of the use of scenarios to address rural poverty alleviation in developing areas [20,55,56], and fewer still in Sub-Saharan Africa [45,47]. This work aims to contribute to this knowledge gap by creating national scenarios with subsets of linked regional scenarios. The resulting scenarios are not linked to global scenarios, which allows for new and independent trajectories to help increase ownership by the participants of the construction workshops.

We worked in Mozambique for a number of reasons: the research team has extensive experience working there; it is relatively politically stable (although in recent years it has seen large policy disruptions); it has high population growth; there are high poverty rates both in rural and urban areas; there are considerable economic development opportunities; small-scale farmers have a high direct relationship with their surrounding ecosystems; and it presents a dynamic mix of environmental risks and LULC changes (see Table 1 for figures about these aspects). Despite great economic development during the 1990s and 2000s, Mozambique still has one of the highest rates of poverty in the world [8]. This is probably due to a combination of the colonial history, two decades of civil war, recurring economic crises and climate related hazards. Agriculture is the main rural livelihood and represents 95% of rural employment and 20% of national GDP [57,58].

The overall objectives of the paper are: (1) to improve the understanding of the complex phenomena linking LULC change and wellbeing of small-scale farmers in sub-Saharan Africa; (2) to identify the driving forces of LULC change; (3) to contribute to the debate about possible futures in Mozambique; and (4) to illustrate the produced scenarios, so that they can be used as sub-Saharan African scenarios in future studies and policy settings.

To reach these objectives, we developed a multi-scale and participatory framework for rethinking land use in Mozambique, highlighting its relationship with human wellbeing. As a result, we developed one set of national scenarios and three sets of regional scenarios for the provinces of Gaza, Zambézia and Niassa (Figure 1). We developed the scenarios in 2015 to support building models connecting land use, ecosystem services and poverty alleviation. Previous studies revealed that biodiversity and poverty were inversely related, suggesting that poverty reduction would imply loss of biological diversity [59]. The relevance of the analysis of these scenarios is that these will help to improve the understanding of the complex phenomena linking LULC change and poverty reduction, which can support current policy decisions such as, for example, the preparation of the Nationally Determined Contributions under the United Nations Framework Convention on Climate Change, land use planning decisions or rural poverty programmes.

**Table 1.** Social and environmental facts and figures from Mozambique that help to frame the scenarios exercise and result.

| Factor | Figures (Year) | Data Source |
|---|---|---|
| Economic Growth | GDP: (Mill USD) 2012: 11,608; 2018: 14,457<br>GDP per Capita (USD): 2012: 607; 2018: 490 | [60,61] |
| State Budget | 5637 Mill USD (2019) | [62] |
| Tourism | 3.4% of GDP, 2.8% of total employment (2017) | [63] |
| Forest area | 47% of the country has some kind of forest cover (34 million ha) (2016) | [64] |
| Agricultural technology | <10% of farms use improved seeds, <5% of farms use fertilizers and <10% of farms use animal traction (2015) | [65] |

**Table 1.** *Cont.*

| Factor | Figures (Year) | | | | Data Source |
|---|---|---|---|---|---|
| Farming commercialization | Less than 20% of rural households sell their produce (TIA 2007). | | | | [66] |
| Climate change | Increase of 1.5/3C in 2046–2065; Changes in raining patterns; Decrease 20% of agricultural production; Increase in extreme events. (2009) | | | | [67] |
| Vulnerability to climate change | 36% of farmers lost part of their crops because of droughts, 30% of farmers lost part of their crops because of floods | | | | [65] |
| | | Mozambique | Niassa | Zambézia | Gaza | |
| Population | 2017 | 27.9 million | 1.7 million | 5.2 million | 1.4 million | [68] |
| | Projected for 2035 | 43.8 million | 3.2 million | 8.1 million | 1.6 million | [69] |
| Urban population | 2014 | 21% | 26% | 21% | 26% | [69] |
| | Projected for 2035 | 36% | 26% | 36% | 26% | [69] |
| Population below 25 years old | 2017 | 66% | 69% | 69% | 64% | [69] |
| | Projected for 2035 | 60% | 62% | 65% | 54% | [69] |

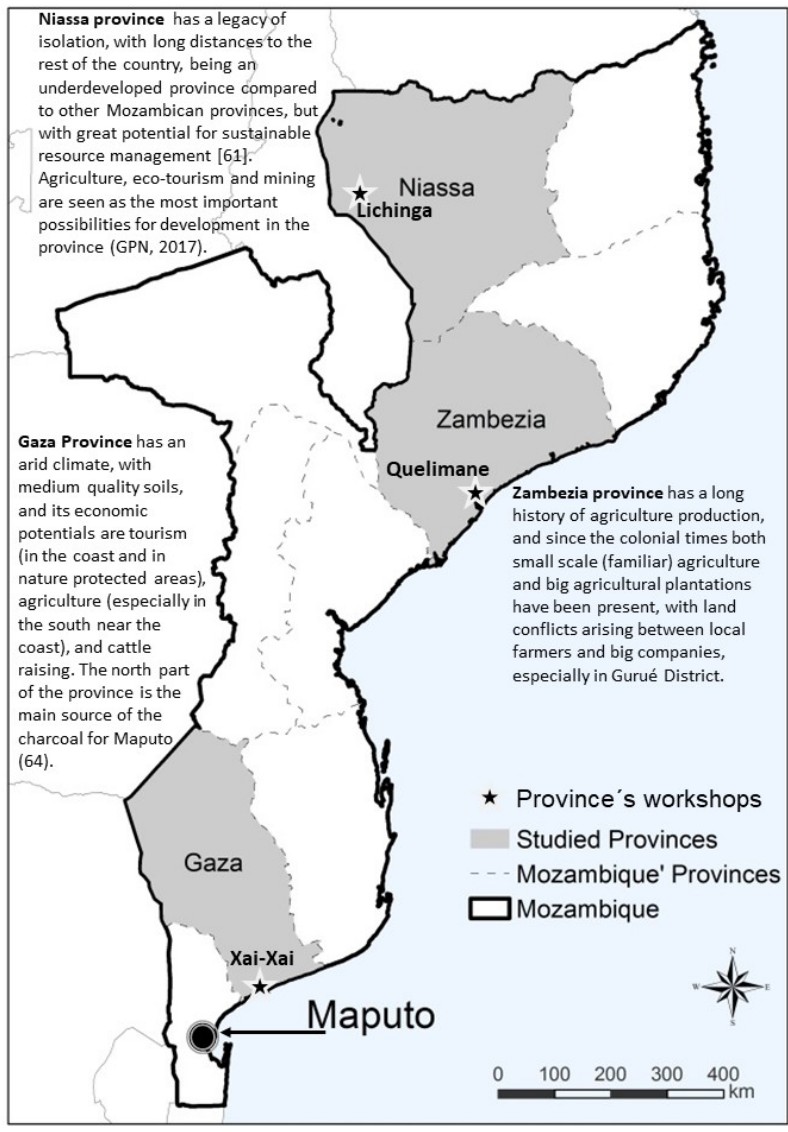

**Figure 1.** Map of Mozambique, with the provinces where the scenarios have been downscaled highlighted in grey colour. Sources of information: [67,70]. More relevant information about each province can be found in Supplementary S1.

## 2. Methods

### *2.1. Study Area*

In order to have the most representative outputs possible, three contrasting provinces in the North, centre and the South of Mozambique were chosen. These provinces differ with contrasting land use transitions and pressures, as explained in Figure 1 and with more detail in Supplementary S1.

### *2.2. Multi-Scale Participatory Scenario Planning Approach*

We followed a six-step approach (Figure 2), based on Metzger et al. [33]. A detailed description of scenario development methodology is included in Supplementary S2, and a summarized description below.

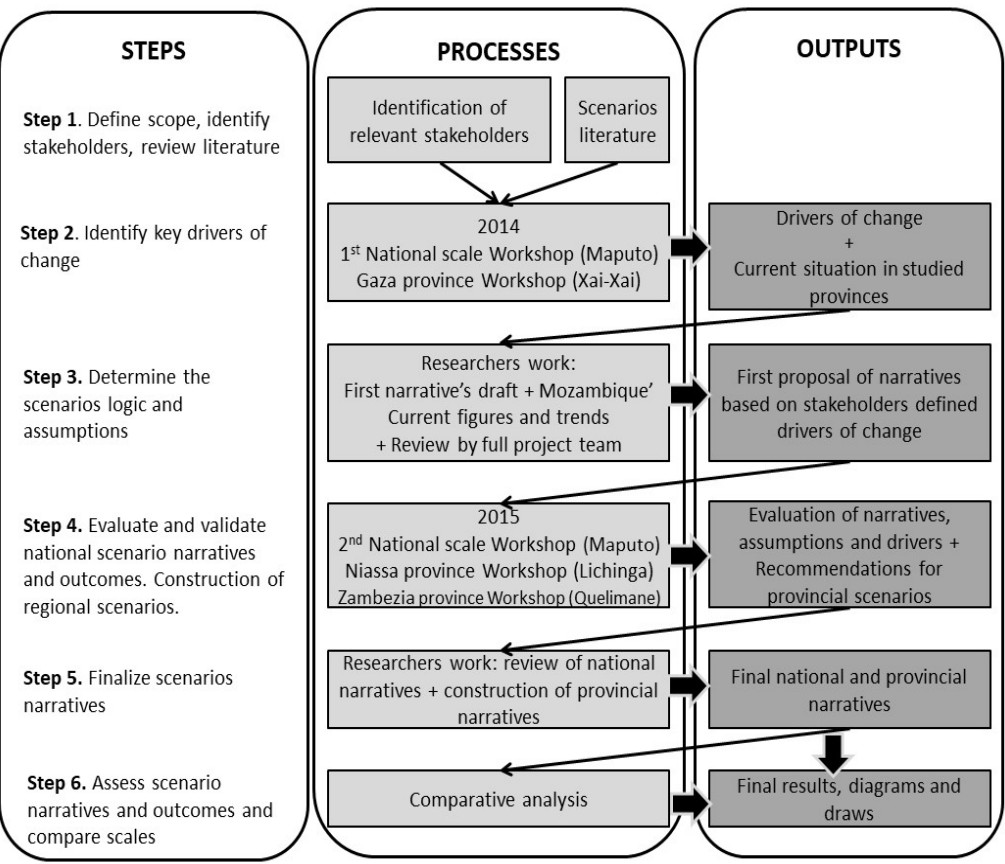

**Figure 2.** Methodological steps followed in the research, building on the methodology presented in Metzger et al. [33].

#### 2.2.1. Step 1. Define Scope, Identify Stakeholders, Review Literature

The first stage was devoted to defining the goals and desired outcomes for scenario construction. This phase included reviewing previous scenario exercises in Mozambique and in neighbouring countries [55,71–73], and identifying a preliminary list of relevant drivers of change for land use, ecosystem services and human well-being.

A stakeholder analysis [74] identified the different types of Mozambican stakeholders from public, private, non-governmental organizations and academic institutions working in rural development, finance and management, environmental management, energy, agriculture, forestry, livestock and tourism.

#### 2.2.2. Step 2. Identify Key Drivers of Change

A first round of workshops consisted of a workshop with stakeholders working with institutions at a national level (Maputo, August 2014: 23 participants); and one with

stakeholders working at provincial and district levels working in the province of Gaza (Xai-Xai, August 2014: 14 participants; see Table 2 and Supplementary S3 for more details). The time-frame of the scenarios was defined through a consensus agreement.

**Table 2.** Number of participants representatives from each sector (government, private sector, NGOs and academia) in each of the five workshops developed.

| Location Date Total Number of Participants | Number of Participants Representatives from Each Sector (Government, Private Sector, NGOs and Academia) |
|---|---|
| **Maputo** 12 August 2014 23 participants | 5 participants from ministries (State Administration: Rural development, Agriculture: Environmental Management, Mineral Resources: Mines; 8 from provincial governments (Agriculture, Tourism, Planning and Finance, Rural energy market, and Environmental action), 2 participants from national NGOs; 3 from international NGOs; and 5 from Universities (Agriculture and forestry and Polytechnic). |
| **Xai-Xai (Gaza province)** 14 August 2014 14 participants | 5 participants from the provincial government (Agriculture and Food Security; Forests and wild animals), 6 participants from district government (Economic Activities: the main governmental institution in the district), and 3 participants from local NGOs. |
| **Lichinga (Niassa province)** 4 August 2015 25 participatns | 10 participants from the provincial government (Directorate of Agriculture, Directorate for Gender, Children and Social Action, Service of forests and wild animals, Niassa national reserve, Directorate of rural energy market, Directorate of Tourism), 2 from the district government (Economic Activities: the main governmental institution in the district), 1 from ecotourism, 1 from a private forest and wood processing company, 1 independent consultant, 4 from national-local NGOs, 2 from international NGOs, and 4 participants from universities (Education, Agriculture). |
| **Maputo** 12 August 2015 14 participants | 3 participants from ministries (Wildlife Department; Directorate of Children, Adolescents and Family; Land, Environment and Rural Development), 1 from provincial government (Environmental Coordination), 3 from National NGOs, 1 from an international NGO, 1 from the National Institute of Disaster Management, 5 participants from universities (Agriculture and forestry, Socio-Economic Studies). |
| **Quelimane (Zambezia province)** 28 October 2015 21 participants | 1 participant from the national government (REDD + Technical Unit of Ministry of Land, Environment and Rural Development), 6 participants from the provincial government (Directorate of Science and Technology, Directorate of Environmental Coordination, Services of livestock, Directorate of Land Environment and Development, Directorate of wood resources, Directorate of Economy and Finance), 2 from the district government (Services for Economic Activities, Services Planning and Infrastructure), 3 participants from wood and agricultural companies, 4 from national NGOs, 3 from the university (Marine and Coastal Sciences, University of Zambezia, Polytechnic University), 1 from the Gurué Agricultural and Livestock secondary Institute and 1 from the Mozambique Agricultural Research Institute–Zambézia. |

During the two workshops, participants worked in groups to identify the main drivers of change affecting rural wellbeing, LULC and ecosystem services (check step 4 to see how we avoided missing important drivers because we developed this step at national level and in one province). Drivers of change were derived from participant's thoughts on what produced large transformations in society and the environment. Each group wrote down the key drivers of change structured into five categories: social, political, economic, technology and environment using the Ketso toolkit (© Ketso Ltd. 2018, Manchester, UK, www.ketso.com (accessed on 1 November 2020). The Ketso toolkit provides tags in the form of coloured leaves and branches of different sizes to display participants' contributions on felt mats. Each participant wrote at least one factor for each category, and then the groups continued adding drivers of change for the five categories. The general objective at this stage was to take into account as many drivers as possible to ensure that we did not miss any important driver of change.

Once the group had finished proposing drivers of change, they worked to identify the most important and most uncertain drivers. Each participant added 2 stickers to the most important drivers and 2 stickers to the less important ones and added the votes. Finally,

after an internal discussion, the group agreed the 2 most important drivers (those causing the biggest changes from the current situation). Using the same method, they identified the 2 most uncertain drivers (those with the highest uncertainty about its future development) similar to the method used in Enfors et al. [46]. Those drivers are reflected in Table 3.

**Table 3.** The most important and uncertain drivers of change proposed by the participants during the first round of workshops, as proposed by each working group.

| | | Most Important Drivers of Change | Most Uncertain Drivers of Change |
|---|---|---|---|
| **1st National Workshop** | **Group 1** | • Empowerment of communities in the management of natural resources <br> • Dissemination of laws and elaboration of land-use plans <br> • Establishment of means for punishment (criminalization of adverse environmental impacts) + monitoring of the implementation of projects | • Recreational use of nature by inhabitants <br> • Environmental protection |
| | **Group 2** | • Access to extension services <br> • Effective application of legislation | • Fair prices <br> • Reduction of gold digging (garimpo) and furtive hunting |
| | **Group 3** | • Decentralization and de-concentration with the participation of the civil society <br> • Economic growth and development | • Effective decentralization <br> • Economic development |
| **Gaza province Workshop** | **Group1** | • Social conflicts demanding development actions <br> • More inclusive political decisions | • Improve rural technologies <br> • Decreasing groundwater levels <br> • An economy based on extractive industry |
| | **Group 2** | • Improvement of rural technologies <br> • Improve environmental policies <br> • Improving rural income | • Balances of payment (public deficit) <br> • Importation/Exportation balance <br> • Reforestation <br> • Social protection |
| | **Group 3** | • Rural emigration <br> • Effective application of legislation | • Improve employment opportunities <br> • Erosion increase <br> • The fragility of an effective application of legislation |

### 2.2.3. Step 3. Determine Logic and Assumptions of the Scenarios: Post-Workshop Analysis and Construction of the First Version of Scenario Narratives

The scenarios had an exploratory goal [75] and a descriptive perspective [39]: the goal was to create a range of likely future alternative scenarios to examine plausible futures, therefore each scenario was not directed towards a single outcome (like normative scenarios do), but rather it was focused on exploring a range of plausible futures [40]. We (the research team using the inputs from workshops and from the literature) followed a combination of the "morphological" approach with the "intuitive logics methodology" [39]. The morphological approach visualizes all the possible interrelations between all potential factors, without prejudging the value of any of them [76,77]. Compared to the "two-axis" approach [78], the "morphological" approach is not restricted to two aspects (those that determine the axis), but rather incorporates a combination of different drivers, giving a similar importance to each of them, thus allowing the elaboration of complex and

transparent scenarios. This approach can increase the relevance, coherence, plausibility, and transparency of the future alternative scenarios generated [77].

Following a morphological approach, we (1) clustered the drivers of change proposed by the participants in the five domains identified (following Metzger et al. [33]); and (2) searched for data supporting the current state of the different drivers. This meant we could propose future figures and combinations for these domains (see Supplementary S4: Table S1). From the full range of possible states, the drivers of change could take in the future, and the possible interrelations between them, we then followed the "intuitive logics methodology" [39]. For this, we (1) analysed the drivers and outcomes of the workshops to identify those considered most important by the participants in the workshops; (2) selected the key drivers, that were used to structure the future alternative scenarios; (3) based on the different possible future states of the key drivers, we selected a meaningful and coherent combination of drivers of change, and their possible future states, one for each of the different possible scenarios; (4) we wrote realistic and coherent descriptive narratives to explain the different outcomes of each scenario.

We decided to construct three scenarios; this number provides enough variability, but avoids adding too much complexity.

### 2.2.4. Step 4. Evaluate and Validate National Scenario Narratives and Outcomes. Construction of Regional Scenarios

With the same diversified range of stakeholders described in Step 1, we held a second and final set of workshops at national level in Maputo (October 2015, 14 participants) and at provincial level in Quelimane (Zambézia Province, October 2015, 21 participants) and Lichinga (Niassa Province, October 2015, 25 participants). The objectives of the national workshop were to evaluate the first version of the scenarios and to refine them to create a final version. Similarly, the provincial workshops evaluated the first version of the scenarios and produced a more refined version of provincial scenarios.

The three workshops followed the same process: participants were divided in five groups and each group worked with one thematic area (social, environmental, political, economic or technological). Each thematic group had to respond to the next set of 4 questions: plausibility of each scenario, whether one factor needed more attention, if any important driver or aspect was missing, and whether any driver should be taken out because of its low importance. Finally, each group explained to the other groups their results and a discussion followed.

### 2.2.5. Step 5. Finalize Scenarios Narratives

The narratives of the national scenarios were updated to incorporate the inputs from the second round of workshops (e.g., including a new driver of change or changing the assumptions in some of them). The scenarios were originally developed at the national scale, and then modified with the input of the provincial workshops to represent the contexts of each of the provinces. Comments from the participants about provincial-specific aspects were used to develop the provincial scenarios, adapting the national narratives to the provincial realities.

### 2.2.6. Step 6. Comparison of Provincial Scenario Narratives

Finally, the results for each province and for the national scenarios were compared and analysed. The main problems were identified and the agreed policies proposed in each province were compared, and differences and commonalities highlighted. The sequential elements of the narratives in each scenario were compared and finessed to ensure that policies were applied in different ways for each of them. The results of the comparison have been included in the discussion section.

## 2.3. Comparison of Scenarios Narratives with Actual Pathways

We compared the constructed scenario narratives with actual pathways in Mozambique since 2015, when the workshops and the narratives were built. The current situation has been matched to the most similar result of each scenario for the main drivers of change.

## 3. Results

### 3.1. Scope of Scenarios (Step 1)

Participants agreed a 20-year time-horizon for the development of the scenarios set for the year 2035, which aligns with the Mozambique National Development Strategy [60] developed for the period 2015–2035 (and operationalized through the government planning cycles and political agendas developed every five years).

### 3.2. Definition of Drivers of Change (Step 2)

The most important and uncertain drivers of change proposed by the participants in the first round of workshops (at National level and in Gaza province) are included in Table 3. The key drivers of change used to build the scenarios were: (a) Empowerment of communities and civil society, (b) Effective application of legislation, (c) Decentralization and a higher involvement of society in politics (e.g., more inclusive decision making), (d) Changes in rural technologies (both in communication and agriculture), (e) Economic growth and development, and (f) Migration. (Full data can be consulted in [79].

### 3.3. National Scenarios (Steps 3, 4 and 5)

The three scenarios represent different potential outcomes of LULC change and rural wellbeing in Mozambique for the year 2035 (Figures 3 and 4). Supplementary S4 contains full narratives of the scenarios.

**Scenario A: Large private investments**

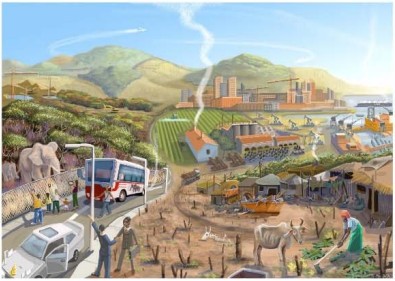

Policies promote international and large-scale private sector as the main development motor; reduced local voice; low implementation of social and environmental policies; globalized approach to resource management. Capital investment increases Mozambique's GDP but equity in society declines, and most rural communities do not improve their livelihoods.

**Scenario B: Small holder promotion**

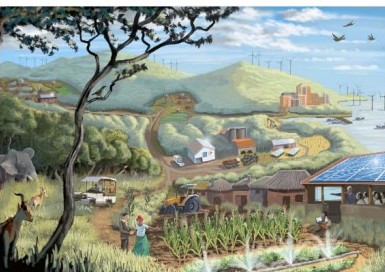

Social and environmental policies are successfully applied, in part because a demand from society: the proliferation of internet-based technologies, also in rural areas, increases the voice of local organizations and a more open and transparent governance model. Improved education and training in rural areas contributes to rural communities improving livelihoods.

**Scenario C: Intermediate**

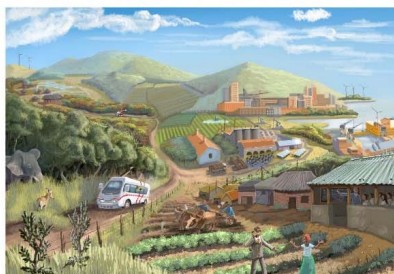

Large private concessions increase, as well as education, with geographic differences; Internet-based technologies enable better democracy; Government economic resources increase from taxes from extractive projects. Some rural communities benefit from large commercial projects and others from improved social services; however, food security remains as the main concern for many communities.

| Themes | Drivers | Scenario A | | Scenario B | | Scenario C | |
|---|---|---|---|---|---|---|---|
| **Politics** | Social and environmental policy implementation | Weak | ↓ | Strongly improved | ↑ | Improved | ↑ |
| **Economy** | Rural family incomes | Low | ↓ | Improved | ↑ | Some improved | ↑ |
| | National GDP | High | ↑ | Medium | ⇒ | High | ↑ |
| **Technology** | Agricultural mechanization | In large projects | ↑ | In local projects | ↑ | Both | ↑ |
| | Agricultural practices | High input | ↑ | Conservation | ↑ | Both | ↑ |
| **Environment** | Forest cover | High decrease | ↓ | Low decrease | ⇒ | Medium decrease | ⇒ |
| | Extraction industries | Intensive | ↓ | Sustainable | ↑ | Intensive | ↓ |
| **Society** | Social services | Same as current | ⇒ | Improved | ↑ | Improved | ↑ |
| | Employment | Only in large projects | ↓ | Med. in urban; High in rural | ↑ | Slight increase | ⇒ |

**Figure 3.** Designs, summaries, and key drivers of change describing respectively the national scale scenarios constructed via a participatory process for Mozambique in 2035. Data supporting the drivers of change can be consulted in Supplementary S4. Designs from the three scenarios by "Ross MacRae".

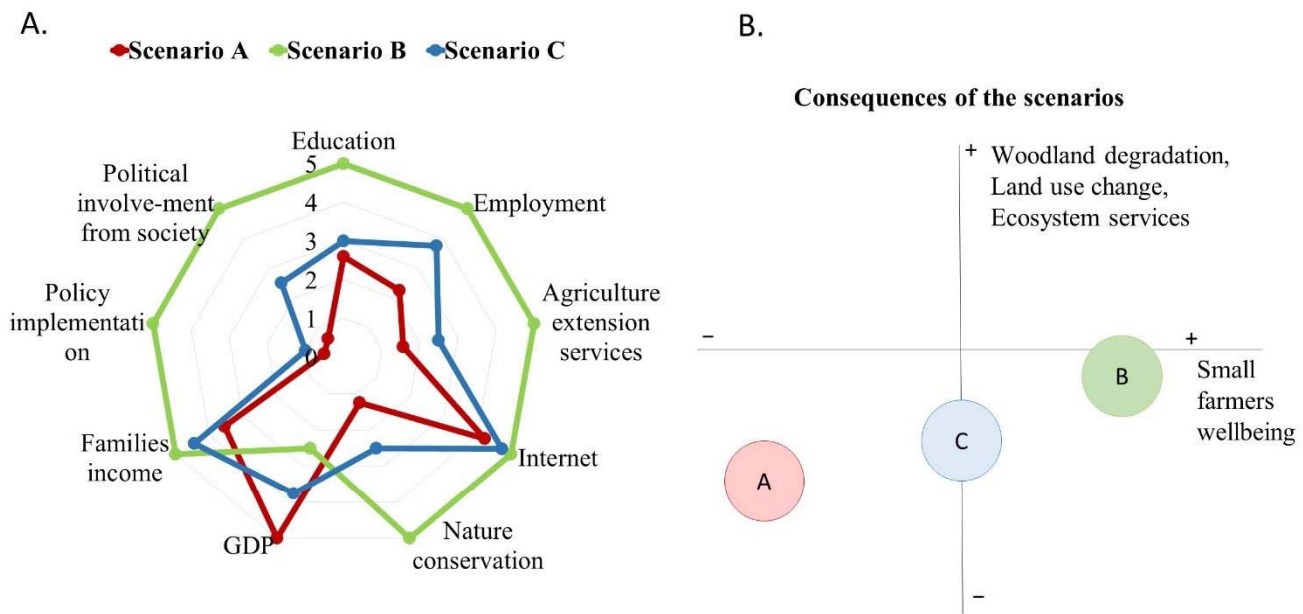

| | Social | | Technology | | Environment | Economic | | Policy and politics | |
|---|---|---|---|---|---|---|---|---|---|
| | Education | Employment | Agriculture extension services | Internet | Nature conservation | GDP | Families income | Policy implementation | Political involve-ment from society |
| Scenario A | 2.6 | 2.3 | 1.6 | 4.2 | 1.3 | 5.0 | 3.6 | 0.5 | 0.6 |
| Scenario B | 5.0 | 5.0 | 5.0 | 5.0 | 5.0 | 2.5 | 5.0 | 5.0 | 5.0 |
| Scenario C | 3.0 | 3.8 | 2.5 | 4.7 | 2.5 | 3.8 | 4.4 | 1.0 | 2.5 |

**Figure 4.** (**A**) Spider gram representing qualitatively the main differences in each scenario to the different drivers of change. (**B**) Diagram representing the impacts of each scenario on small farmer well-being and the environment. Data supporting figures (**A**,**B**) can be consulted in Supplementary S4 (Tables S1 and S2).

### 3.3.1. Scenario A: Large Private Investments

Scenario A is characterized by public policies that promote international and large-scale private sector as the main development motor, accompanied by low implementation of social and environmental policy provisions. Scenario A presents also a reduced local voice (participation), and adopts a globalized approach to resource management. As a consequence, more of Mozambique's land is under private long-term leases and concessions by 2035. This includes agricultural and forested areas but also a significant increase in mined areas. The government favouring large foreign capital investment, together with an ineffective land use policy and an increase in technological advances, results in higher migration of rural populations to urban areas. Although capital investment considerably increases Mozambique's Gross Domestic Product (GDP), equity in society declines, and most rural communities do not improve their livelihoods (food security is their main concern). Implementation of state-led social and environmental policies is not effective due to lack of funding, e.g., public extension services continue to be scarce. Environmental quality also decreases in many ecosystems as a result of intensive land management. Mozambique's relations with its neighboring countries are improved through greater trading partnerships including China, many European Union countries, Brazil and India. Climate change adaptation and mitigation strategies are more reactive than proactive.

### 3.3.2. Scenario B: Small Holder Promotion

Local power is increased and public policies drive a development agenda focused on promotion and investment in small and medium enterprises. The proliferation of internet-based technologies, also in rural areas, increases the voice of local organizations,

which pushes the government to increase public involvement in rural development and the improvement of public services. This scenario assumes there is also a real commitment from the government to improve education and training, and a more open and transparent governance approach. Social and environmental policies (e.g., education and training, health, water, extension services, and protected areas) are a priority for the government, partly due to societal demand in tandem with NGOs. Most rural communities improve their livelihoods: food sovereignty is achieved due to a sustainable and small-scale agriculture production, with a focus on extension services. Public support to communities results in sustainable forest management, which seeks to protect plant and animal diversity through harvest levels that respect ecosystem integrity. There are many areas for protected wildlife, and some are used for community-controlled eco-tourism. Mozambique welcomes international investments based on the requirement that companies respect local communities and share the development profits. Climate change adaptation strategies are strategically applied in small projects rather than in large programs.

### 3.3.3. Scenario C: Intermediate Scenario

This scenario presents a balance between a more globalized approach versus one with regional and local community empowerment in resource management. Large parts of Mozambique's land are in long-term private leases or concessions. However, an improvement in education, empowerment, and environmental stewardship allows some communities to self-organise and improve their well-being. Internet-based technologies enable better democracy and allow community empowerment to flourish in some areas of Mozambique, although the state still maintains a high control of resources and power. The economic government resources are higher because a greater percentage of income from taxes is levied on international extractive projects. This has special importance in some districts that have improved public services and community empowerment. Some rural communities benefit from large commercial projects, whereas other communities benefit from the improvement of social services. However, food security continues to be the main concern for the rest of the communities. There are several areas of protected wildlife, yet environmental quality decreases in many habitats and ecosystems as a result of intense use of resources. Climate change adaptation strategies are strategically applied in small projects rather than in large programmes, and there is an improvement in awareness raising, education and investment capacities.

The trends of land use land cover change under each future scenario are as follows:

1. Under Scenario A "large private investment", deforestation is driven by large companies that transform large parts of the country into agricultural land and achieve high rates of mining and timber extractions; urbanization is driven by rural migration to urban areas, in part due to the loss of land due to exploitation from private companies; woodland degradation is driven by the charcoal demand from the new urban inhabitants.
2. Under Scenario B "small holder promotion" rural families have a larger role and more power in decision-making resulting in agricultural land expanding into forests, more farm extension services and a growth of medium size farms. This scenario also assumes the government increases its capacity to enforce laws for the protection of natural areas. Nevertheless, the development of small scale farming and the increase in medium scale farming around the country also results in an increase in deforestation in non-protected areas.
3. Under Scenario C "Intermediate" both paths take place with similar intensity: agricultural expansion from small farmers, woodland degradation from charcoal demand and natural area degradation due to the impact of large investments in agriculture, mining and timber extraction. Nature protection is better achieved than in scenario A.

### 3.4. Province Scenarios (Step 6)

The downscaling of the national scenarios to the three provinces of Niassa, Zambézia and Gaza produced parallel scenarios with specifics in each of them (Table 4).

**Table 4.** Comparison of the scenarios in each Province.

| | Niassa Province | Zambézia Province | Gaza Province |
|---|---|---|---|
| **Introduction** | The three scenarios imply a large expansion of infrastructures (roads and train connections) to facilitate agricultural expansion and transport of products. Interventions most voted by participants include: (a) the promotion of farmer's associations and (b) promotion of community natural resource management. | The interventions most voted by participants in the workshop include: (a) improving law compliance, (b) improving the transfer of agrarian technology to farmers to encourage conservation agriculture, (c) land use planning; (d) facilitating the process of acquiring land rights by farmers and the delimitation of communal areas. | The environmental consequences of charcoal production are a big concern, even if agriculture is the key economic activity. Proposed interventions: (a) to improve agricultural extension services and other agricultural services to increase farm mechanization and irrigation; (b) to improve the use of better seeds; (c) to promote alternative energy sources and improved charcoal stoves for urban consumers; and (d) to increase capacity building of rural communities. |
| **Sub-Scenario A** | Increase in the level of industrial activity, especially in mining operations. An increase in oil production in Niassa Lake opens a dispute in the Rovuma Basin between Malawi, Tanzania and Mozambique. Illegal timber operations grow due to the difficulties to obtain legal permits. PROSAVANA development project benefits especially big agricultural firms, producing the displacement of a large population to worse lands. | The government promotes large private agricultural schemes. Implementation of social policies is a challenge due to the number of private companies involved and a weak government capacity to enforce laws. Many farmers are moved from their lands, land conflicts increase between investors and smallholders, and a big part of the population migrates to other provinces, to cities, and to other countries. | The proposed interventions are not effectively applied by the government that is more focused on facilitating the implementation of large plantations, which occur mostly in the best agricultural land. Urban charcoal demand increases greatly, as a result of the great migration to urban centres, in part because of the problematic situation in the rural areas (see other provinces). |
| **Sub-Scenario B** | Big firms give up agriculture and forest plantations because of problems with bureaucracy. The government is successful in the promotion of irrigated agriculture, with big, medium and small infrastructures that allow farmers associations to increase their productions notably. An increase in tax revenues allows more access to credit by small farmers and more diversified job opportunities with most families improving their livelihood and wellbeing. Successful promotion of sustainable agriculture to small farmers, moving a high proportion of them out of poverty. PROSAVANA development project is directed to benefit small and medium scale farmers. | The promotion of conservation agriculture is successful (following an existing example by the NGO CLUSA). The government promotes small companies with public procurement procedures, like for small artisans and factories making pavements. In 2035 small companies are producing as a family sector. Improvement of access to IT in rural areas at accessible costs is achieved (as a combination of efforts from the government, NGOs, private companies and farmers). The use of solar panels increases (examples already exist in the province). Farmer movements obtain investments from the government and international bodies to improve water infrastructure in the Zambezi river, which increases agricultural production, especially for staple crops like rice. | The proposed interventions are applied successfully, since the government seeks to improve local rural capacities and nature protection. Urban charcoal demand remains constant, a result of low migration from rural areas to urban centres and an increase in the use of other types of energy, like renewable energies, that are promoted by the government and international organizations. |
| **Sub-Scenario C** | Reasonable investments in industrial development and more consciousness by taxpayers. PROSAVANA produces the displacement of some farmers, with others benefitting from the new infrastructure built, from private extension services and from a new variety of crops' value chain. | The problems from an informal style of doing things influence big investors, with a large part refusing to invest in Zambézia. Expansion of the Emergent Farmers' model: a greater proportion of land is controlled by medium size farmers (farming between 20 and 50 ha) increasing the production of horticulture and livestock and improving soil management. | Charcoal demand increases, but not as much as in Scenario A. Some interventions are successfully applied, especially those related to access to technologies, which facilitates communication for the population, who demand and achieve a substantially improved capacity for self-organization. |

There are also differences in the main policies implemented, which reply to the different needs in each province. For example, in Niassa important policies imply the expansion of transportation infrastructures and the implementation of the PROSAVANA project, a large-scale national project directed towards the development of agriculture. In Zambezia, the improvement of extension services was detected as a crucial policy, together with improving the land tenure situation, which is critical nowadays because of the relations between large-scale investments and local farmers. In Mabalane, agriculture development needs improvements of infrastructure such as irrigation and improvements of the charcoal production.

*3.5. Comparison of Scenarios Narratives with Actual Pathways*

Since 2015, when the last workshops took place, Mozambique has taken a trajectory that aligns more with scenario A, but also partially with Scenario C (due to territorial differences) (Table 5). Since the scenarios were designed in 2015, important development landmarks have characterized the country. Particularly, the hidden debt crisis which arose in 2016, and slowed economic growth from about 7% to less than 4% and the Idai and Kenneth cyclones in 2019, which further reduced the economic growth to about 2%. It is expected that the COVID-19 pandemic will reduce the economic growth even more. This highlights the importance of aligning different interventions and following an integrative or systemic perspective, as individual initiatives can fail if they are not supported by complementary investments like infrastructure, social services or markets [46].

**Table 5.** Brief description of the Mozambican trajectory since 2015 and its correspondence with scenarios A, B and C.

| Mozambican Trajectory Since 2015 | Correspondence with Scenarios |
| --- | --- |
| The number of mine concessions has increased, increasing the power of large companies, and producing some negative effects on local farmers (i.e., conflicts in Cabo Delgado). | In line with Scenario A |
| The oil and gas sector took important measures with the final investments decision totaling more than 50 billion USD investment between 2017 and 2019 by multinational groups. This would have allowed an increase in social policies supporting small farmers. Nevertheless, due to the decrease in other sums (especially cuts in aid to governments by donors), social spending decreased. | In line with Scenario A (no increased support to small farmers). |
| Meanwhile, some areas benefitted from small and medium scale agricultural and forestry projects, e.g., a project funded by the World Bank (SUSTENTA project), in Zambézia and Cabo Delgado provinces, FAO projects and a Sweden supported project in Niassa province [80]. | In line with Scenario C (territorial differences, with some areas benefitting and others not doing so). |
| Internet connections has not increased as in earlier periods. This is one of the drivers of change of the scenarios: Scenario B assumes there is a great increase in access to internet connections, which results in a higher civil society organization. | In line with Scenario A. |

## 4. Discussion

### 4.1. Understand the Complex Processes That Link Land Use Change, Nature Degradation, and Poverty Alleviation

The resulting scenarios can be considered more adaptive than transformative [81], because they were designed more to look for interventions that could increase social and environmental sustainability under each different scenario than to look for potential sustainable futures. This has allowed a better understanding of the complex relationships between LULC change and local populations' well-being. Participants agreed that the main direct drivers of LULC change in Mozambique are the increase in agricultural land, urbanization, deforestation due to extractive activities like mining and timber production, and land degradation due to firewood and charcoal production (see full narratives in Supplementary S4). Under the different scenarios constructed, all these trends continue, but at different rates, patterns, and origins depending on the specific drivers and on the complex relationship between those drivers. Previous research indicated that a bottom-up



or participatory resource governance would imply higher nature conservation results [82]. Although some participants of the presented research also agreed with this view, this was not totally agreed by all of them because bottom-up driven scenarios (in this case, Scenario B) can also head to high forest degradation and deforestation.

The scenarios show several complex interactions between drivers. For example, changes in rural technologies like small-scale solar panels and new communication technologies can allow the development of small-scale farms with a path that is less government-dependent compared to the current situation. Access to electricity with solar panels allows access to many other technologies (mobile phones, radio, refrigerators, TV, etc.) and changes in habits and social behaviour (e.g., enabling night-time study). The deployment of IT allows access to a wide knowledge repository stored in the web and facilitates communication and organization of civil society. This could be used both for increasing the demands pitched to the government and a better self-organization of communities. This is a key factor for development and poverty reduction. For example, stakeholders in Ethiopia considered that participatory forest management was useful to increase forest income in the long-term [47]. Nevertheless, the same study highlighted the difficulties faced by participatory initiatives due to weak accountability and growing inequalities or problems for controlling management decisions. More recent research also in Ethiopia recognized that participatory resource governance, and local agency would contribute to increasing natural capital and provide diverse harvests [82]. In our work, stakeholders proposed that the increase in societal leadership would allow civil society to push the government for improving social services. Scenario B is characterized by this process: an improved access to the internet in rural areas contributes to a significant improvement in education and extension services for small-scale farmers. This was identified as critical for a scenario of sustainable farmers' development in two research scenarios in Tanzania [46,48] and that previously referred to in Ethiopia [82].

Another example of the complex links concerns the effective application of the law. It was highlighted by most participants in the workshops with the common assertion that "Mozambique has good laws and plans that are not applied". Nevertheless, instigating effective application of the law and planning would result in important changes and very different futures depending on the government priorities. Participants commented that in the current situation, many large (mining, forestry and agriculture) companies are not investing in the country due partly to the poor application of the law, which decreases investors' confidence and certainty. Effective application of the rule of law will increase foreign investments in large projects, although in some cases this could increase conflicts with local inhabitants and decrease local well-being [83–85]. If these large projects were to occur, they would result in higher national tax revenues, which could potentially provide additional resources for improving social services. At the same time, more stringent application of the law could increase social services, and therefore increasing small-scale farmers' well-being.

Participants in the workshops highlighted the importance of peace as an important political factor in Mozambique. However, we did not include war as an option (i.e., a "shock event scenario"), because this would imply that any planned policy could not be implemented [71], and therefore it would have a small interest for policy makers. These events suggest that 'shock event' scenarios should be implemented in future projects, and methods like the OLDFAR algorithm [86] could be used in future scenario developments to achieve an optimally diverse set of scenarios.

### 4.2. Assessing the Multi-Scale Approach

Our scenarios are not embedded within global scenarios (c.f. [36]), but start from the analysis of national and regional driving forces. The method followed in our research allowed us to define the main driving forces for national and provincial levels simultaneously, involving national and local stakeholders. The process also allowed us to include regional and local perspectives in the national narratives enabling links between scenarios

across different scales. Although the scenarios were first constructed at a national scale and then downscaled to regional areas, stakeholders from the provinces also proposed useful considerations for the national narratives. Their opinions were used both to evaluate the national narratives and to downscale them to the provincial scale. The involvement of stakeholders from the provinces provided knowledge of local realities, which was essential to root the scenarios in the reality of the country [82]. Following the framework set by Zurek and Henrichs [87] the scenarios presented should be classified as "consistent across scales": the regional scenarios share clear boundary conditions but each of them present different outcomes depending on the regional reality. The mixed method presented, by which participants contribute to the national scenarios and to configure provincial scenarios does limit the variability between provincial scenarios [17]. Nevertheless, the inclusion of local and provincial factors in the downscaling exercise allowed us to differentiate between provincial scenarios [88]. "Consistent across scales" scenarios are defined as useful for linking and comparing scenarios across regions [55], in line with Biggs et al. [44], who find the existence of loose links useful because they help maintain credibility and allow specific differences.

The multi-scale approach has highlighted the different consequences of scenarios in each scale and for each province. Across provinces, scenario A has the same impact: a decrease in natural areas, but due to different root causes (charcoal in Gaza, cropland in Zambezia and mines in Niassa). Another similarity across the three provinces is the vulnerability of small farmers, although they face different threats and opportunities in each province.

The differences between provinces arise from the different realities in each location: in Niassa mining activities and large forestry and agriculture companies have a contrasting effect on country revenues and ultimately GDP, but have direct negative effects on local farmers. In Niassa and Zambézia, the evolution of the ProSavana project is a good example of how quickly policies can change and curiously represents two contrasting elements of our scenario exercise. The original conception was to encourage industrial agriculture at large-scales but it has evolved to focus more on promoting small farmers due to public pressure. The real execution of this project still needs to happen. In Gaza, the third province, the northern districts will continue to be impacted by the urban demand of charcoal and could turn around their challenging situation due to droughts by an improvement in water infrastructure, which could also benefit the southern districts, that have higher farming and tourism potential. The participants in the provincial workshops evaluated the drivers and narratives proposed at national scales, and the final scenarios and narratives were adjusted to that evaluation. Therefore, our downscaling exercise allows us to describe more nuanced scenarios, with clear and precise examples of the consequences of the different plausible futures. In the presented case study, the downscaling has highlighted the importance of public policies to deal with external and internal driving forces.

The inclusion of quantified consequences of the three scenarios (e.g., future land use change in percentages, or future changes of regional poverty rates) was very challenging [89]. Reasons for this difficulty were the complex relations between drivers of change, the qualitative focus of the work developed, the lack of expertise from all participants concerning all aspects of the process (they were experts in just one specific sector), and the lack of time to work with rigorous quantitative results. Due to these circumstances, the results of quantifying land use change under each scenario contained contradictions and land use change rates much higher than the widely accepted ones. This learning implies specific time must be devoted to producing quantified results, and this process can benefit from an iterative process using modelling tools [89].

### 4.3. Policy Recommendations

The three scenarios show the difficulties the government face in improving the livelihoods of subsistence farmers. In order to achieve this, the government should promote agriculture extension services to tackle small-scale farmer productivity as well as increase

farmlands size [90,91], and secure land tenure rights for small-holders [92]. However, the outputs of Scenario B suggest an increase in small-scale farmers' productivity and farm area could also increase deforestation and forest degradation [12]. In the face of a likely decrease in natural land cover in the next few decades, achieving effective protection of natural areas will be critical. Scenario A highlights the difficulty of controlling the actions of large companies, and the possible negative impact on small-scale farmers. Concentrating efforts to improve conservation of protected areas in the country [93] must involve local communities to ensure they also obtain benefits from nature protection [94]. Additional economic resources are needed for nature protection and management [14,95] and for impacted local populations. Part of those funds could be obtained from valuing the contributions ecosystems provide society, such as carbon sequestration or natural hazard regulation. Payments for Ecosystem Services schemes could be applied based on the lessons learned from the REDD+ programme. Additional revenues could also come from nature-tourism.

Previous participatory scenario construction processes have proved useful to support governance [89]. The participatory process presented in this paper and the project that developed it has influenced policy making in Mozambique already. Actions to influence policy include the publication of a policy brief and the presentation of the scenarios constructed in a final project conference. The scenarios have also been used to build Bayesian Belief Networks to model the consequences of different policies [12] and to produce maps representing land use change and ecosystem services distribution under each scenario in the province of Gaza [96]. Furthermore, participants in the workshops have been involved in the elaboration of public policies, and the research team has been consulted in those cases, resulting in conclusions from the research project being included in the policies. For example, Mozambique's forest policy has been reviewed and approved early in 2020 [97], calling for the development of a biomass energy policy as the basis for the promotion of sustainable charcoal production. The first NDC were submitted to the UNFCCC in 2018, recognizing that agriculture, forests and other LULC sectors have potential to contribute more than 80% of the greenhouse gas emission reduction [98], implying significant changes in the current dynamics of LULC. In addition, local measures have also been taken, such as the improvement of charcoal licensing and monitoring in Mabalane by the Gaza provincial Forest Service. Furthermore, several research activities have also been implemented to help improve understanding of land use dynamics and charcoal production (e.g., [99–101]).

## 5. Conclusions

The richer understanding and gains in context-specific knowledge on LULC and ecosystem services delivery for human well-being is particularly important in areas with populations of vulnerable small-scale farms. We have explored the interlinked consequences of drivers of change and how different these are when mediated through concrete decisions such as social and environmental policies and public extension services. These can have different context-specific and scale-dependent impacts on livelihoods, ecosystem services as well as LULC. We developed three plausible scenarios. Scenario A is characterized by the promotion of large-scale interventions. It highlighted large LULC changes from mining, agriculture and timber interventions resulting in increased migration from rural to urban areas but a negative impact on many rural livelihoods. Scenario B is centred on the promotion of small-scale farming as a result of societal pressures on the government. It would also produce large LULC changes due to the expansion of small and medium scale farmlands but would have the potential to bring about a more autonomous development and greater farmer empowerment. The capacity of the government for improving social services is necessary in this Scenario B. Higher participatory resource governance and local agency can trigger scenario B, which can be facilitated by new technologies like small scale renewable energy production and communication technologies. The middle road of Scenario C showed how large-scale projects linked with an effective application of the law

can increase public resources, which can also be directed to promote the development of small-scale farmers and small private initiatives.

The results from this participatory scenario exercise aimed to support the exploration of options available to decision-makers in Mozambique. Several policy and decision-makers actively participated in the workshops, resulting in the creation of a policy brief. The co-creation process highlighted the value of the vision process and helped illustrate these complex and interlinked consequences by supporting the understanding and knowledge of trade-offs and synergies to improve future decisions.

**Supplementary Materials:** The following are available online at https://www.mdpi.com/article/10.3390/su132313030/s1, Supplementary S1: Description of the three studied provinces, Supplementary S2: Methodology followed in the development of the scenarios workshops, Supplementary S3: Detailed description of participants representatives from each sector, Supplementary S4: Full scenario narratives. References [102–115] are cited in the supplementary materials.

**Author Contributions:** Writing—original draft: P.Z.-M. and E.L.-M.; Writing—review & editing: P.Z.-M., M.J.M., G.P., A.S., S.N.L., J.S.P. and E.L.-G.; Conceptualization: P.Z.-M., M.J.M., M.M., G.P., J.S.P. and E.L.-M.; Data curation: P.Z.-M., M.M., A.S. and S.N.L.; Formal analysis: P.Z.-M., J.S.P., M.J.M. and E.L.-M.; Funding acquisition: M.J.M. and G.P.; Methodology: M.J.M., P.Z.-M., J.S.P. and M.M.; Validation: A.S., S.N.L. and E.L.-G. All authors have read and agreed to the published version of the manuscript.

**Funding:** This work was funded by the European Union's Horizon 2020 Research and Innovation Programme, under the Marie Sklodowska-Curie Grant, agreement number 798867 and by the ACES project (NE/K010395/1) from the Ecosystem Services for Poverty Alleviation (ESPA) programme (funded by the Department for International Development (DFID), the Economic and Social Research Council (ESRC) and the Natural Environment Research Council (NERC)).

**Institutional Review Board Statement:** Not applicable.

**Informed Consent Statement:** Informed consent was obtained from all subjects involved in the study.

**Data Availability Statement:** Data has been published in the referred data repository. Zorrilla-Miras, P., Matediane, J., Mahamane, M.; Nhantumbo, I.; Varela, R.; Metzger, M.J.; Patenaude, G. (2018). Scenarios of future land use change in Mozambique (2014 and 2015). NERC Environmental Information Data Centre. https://doi.org/10.5285/97c65c35-1db5-49d5-8ee0-ae5c7b699634 (accessed on 20 January 2019).

**Acknowledgments:** This paper is dedicated in memory of Raul Varela, who had a very active role in the development of the workshops that allowed the elaboration of this publication. Special thanks to Isilda Nhantumbo, who had a key role in the development of the conceptual elaboration of the scenarios and in the organization, facilitation and organization of the workshops. Special thanks to Maria Julieta Matediane for her relevant work in the stakeholder identification and in the logistical organization of the workshops, and to all the ACES team.

**Conflicts of Interest:** The authors declare no conflict of interest.

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
