# Peer review of "Understanding Complex Relationships between Human Well-Being and Land Use Change in Mozambique Using a Multi-Scale Participatory Scenario Planning Process"

_sustainability, doi:10.3390/su132313030_

Round 1

Reviewer 1 Report

Respected Authors,

I would like to congratulate you all on bringing together such valuable work, for science and society. Eventually, policymakers could also benefit from it. The manuscript is well written in a comprehensive manner. The methods are clearly explained and the discussion on the three scenarios shows a convincing argument and solid structure.

Please check the coherence in terms of how you titled the scenarios. in the abstract Scenario C appears as an intermediate scenario but later one as Storyline C: Balanced scenario; It is also confusing calling Scenario and Storyline. I would suggest "Scenario" which have their own storylines.

I would recommend to mentioned the projects supporting this research but your Acknowledgments make this clear.

Congratulations

Author Response

Thank you for your comment. We have included now always the word “Scenario”, to avoid confusion with “Storyline”. Also, we have now called always “Intermediate scenario” to Scenario C, to avoid confusion.

Reviewer 2 Report

Original Manuscript ID: Sustainability (ISSN 2071-1050).

Original Article Title: “Understanding complex relationships between human well-being and land use change in Mozambique using a multi-scale participatory scenario planning process “.

This paper is well written. It presents the complex relationships between human well-being and land use change and their possible future evolution in Mozambique up to 2035 using a multi-scale participatory scenario planning process. ​The first scenario is characterized by the government promoting large investments; the second scenario is characterized by the increase of local community power and public policies to promote small and medium enterprises; and the third, intermediate scenario. All three scenarios highlight qualitative large LULC changes, either driven by large companies or by small and medium farmers.

Some comments were written below:

  1. The abstract and conclusion should be more informative, including the main achievement of your proposed study; this should be more like a short quantitative report.
  2. Please use recently published papers in your manuscript.

      3. Please carefully revise and improve the English in your manuscript.

Author Response

Thank you for your comments:

  1. Although the presented work is a qualitative work and not quantitative, we have now included more explicit results in abstract and conclusions.

As the abstract needs to be kept short in number of words, we have included just one more sentence: “The scenarios have different impact in wellbeing and equity, the first one implying a higher rural to urban areas migration.”

More achievements from the work have also included in the “Conclusions” section:

“It would also produce large LULC changes due to the expansion of small and medium scale farmlands but would have the potential to bring about a more autonomous development and greater farmer empowerment. The capacity of the government for improving social services is necessary in this Scenario B. Higher participatory resource governance and local agency can trigger scenario B, what can be facilitated by new technologies like small scale renewable energy production and communication technologies.”

  1. We have also included new recently published papers.

Jiren, T.; Hanspach, J.; Schultner, J.; Fischer, J.; Bergsten, A.; Senbeta, F.; Hylander, K.; Dorresteijn, I. Reconciling Food Security and Biodiversity Conservation: Participatory Scenario Planning in Southwestern Ethiopia. Ecology and Society 2020, 25, doi:10.5751/ES-11681-250324.

Rutting, L.; Vervoort, J.; Mees, H.; Driessen, P. Participatory Scenario Planning and Framing of Social-Ecological Systems: An Analysis of Policy Formulation Processes in Rwanda and Tanzania. Ecology and Society 2021, 26, doi:10.5751/ES-12665-260420.

Allan, A.; Barbour, E.; Nicholls, R.J.; Hutton, C.; Lim, M.; Salehin, M.; Rahman, Md.M. Developing Socio-Ecological Scenarios: A Participatory Process for Engaging Stakeholders. Science of The Total Environment 2022, 807, 150512, doi:10.1016/j.scitotenv.2021.150512.

  1. Finally, the English has been reviewed again by two of our authors that are native English speakers.